# Flower Buds Confirmed in the Early Cretaceous of China

**DOI:** 10.3390/biology13060413

**Published:** 2024-06-04

**Authors:** Weijia Huang, Xin Wang

**Affiliations:** 1Blue Miracle Museum Science Research Studio, Guangzhou 510000, China; bmm2024@126.com; 2State Key Laboratory of Palaeobiology and Stratigraphy, Nanjing Institute of Geology and Palaeontology and CAS Center for Excellence in Life and Paleoenvironment, Chinese Academy of Sciences, 39 East Beijing Road, Nanjing 210008, China

**Keywords:** Lower Cretaceous, China, angiosperms, fossil, flower bud

## Abstract

**Simple Summary:**

Flowers are attractive to many people mainly due to their colourful and conspicuous perianth, which is closely related to successful insect pollination in extant angiosperms. The Lower Cretaceous Yixian Formation (125 million years ago) in Northeastern China is famous for its great diversity of early angiosperms. However, unlike typical flowers in extant angiosperms, the previously documented fossil flowers are usually famous for their “nakedness”, namely, they do not have typical petals (corolla), suggesting a pollination strategy for early angiosperms that is different than that of the extant angiosperms. However, without fossil evidence, whether there are petals (corolla) and whether androecium and gynoecium are protected in early flowers are rarely addressed questions. Here, we document a flower bud fossil, *Archaebuda cretaceae* sp. nov., from the Yixian Formation, which is the second case in the Formation. While reinforcing the truthful occurrence of fragile flower buds in the Early Cretaceous, new fossil evidence demonstrates the existence of gynoecium and possible androecium within the flower bud, suggestive of the possible bisexuality and the protection function of petals in *Archaebuda*, just as in extant flowers. In addition, it is very likely that conspicuous petals may have played an important role in attracting insects for successful pollination in early angiosperms. Therefore, the occurrence of petals (corolla) in *Archaebuda cretaceae* sheds a new light on the reproduction of early angiosperms.

**Abstract:**

The Yixian Formation (Lower Cretaceous) in China is famous worldwide for its fossils of early angiosperms, but there has been only one record of flower buds (*Archaebuda lingyuanensis*) hitherto, in which only the surface of the flower bud was documented while no internal details were known. Such a partial knowledge of flower buds hinders our understanding of the evolution of flowers, and this knowledge lacuna needs to be filled. Our new specimen was collected from an outcrop of the Yixian Formation (Barremian–Aptian, Lower Cretaceous) near Dawangzhangzi, Lingyuan, Liaoning, China. Our observations reveal a new fossil flower bud, *Archaebuda cretaceae* sp. nov., from the Lower Cretaceous of China. This new record of *Archaebuda* in the Yixian Formation not only confirms the truthful existence of the expected gynoecium (plus possible androecium) in a flower bud but also underscores the occurrence of typical flowers in the Early Cretaceous. This new information adds first-hand data to flower sexuality, pollination, and evolution.

## 1. Introduction

The origin and early evolution of angiosperms have been the foci of botanical studies for a long time [1,2,3,4,5,6,7,8,9,10,11,12,13,14,15,16,17,18,19,20,21,22,23,24]. From the Early Cretaceous Yixian Formation in western Liaoning, China, numerous angiosperms (including reproductive organs) have been reported, including *Chaoyangia* [11], *Archaefructus* [12,16], and *Sinocarpus* [15] (to name a few). Besides these reports, various angiosperms have been reported from South America [25,26,27,28,29,30,31,32]. Despite such a great abundance and diversity of early angiosperms [11,12,14,15,16,22,25,26,27,28,29,30,31,32,33]), up to now, there has been only one report of flower bud in the Jurassic [34] and one in the Early Cretaceous [35], respectively. Such scarce information on flower buds underscores the mystery of the history of flowers and makes concerned fossils and studies badly needed. *Florigerminis jurassica* from the Middle–Late Jurassic of China [34] is a flower bud physically connected to leafy branches and fruit, while *Archaebuda lingyuanensis* is a big flower bud lacking information of the internal fertile parts from the Early Cretaceous of China [35]. The common shortcomings of these reports on fossil flower buds are that internal details (the fertile parts) in both cases are missing. Therefore, palaeobotanists are eager to know what is inside the assumed flower bud, and information on these parts can either negate or confirm the flower bud identity of these fossils. Here, we report the third flower bud fossil in the plant history (the second one from the Yixian Formation, Lower Cretaceous). In line with the preceding record, a new specimen of *Archaebuda* with internal details missing in the preceding cases underscores the truthful occurrence of flower buds in the Yixian Formation and provides first-hand information on the gynoecium, possible androecium and corolla in early flowers. The discovery not only adds to the existing great diversity of angiosperms in the Early Cretaceous but also sheds new light on the evolution of flowers in early angiosperms.

## 2. Materials and Methods

Various fossil animals and plants have been documented in the Yixian Formation of Northeastern China [11,12,14,15,16,33,35,36,37,38,39], and the age of their formation used to be a focus of studies for many stratigraphers [40,41,42,43,44,45,46,47,48,49]. Now, there is a general consensus over the age of the formation, namely, approximately 125 Ma (Barremian–Aptian, Lower Cretaceous) [50]. The present fossil specimen was collected from the Dawangzhangzi outcrop of the Yixian Formation near Lingyuan, Liaoning, China (Figure 1). The specimen was preserved as a flattened compression embedded in thin-layered siltstone, although the original organic materials were largely missing (Figure 2a). The whole specimen was imaged using a Sony ILCE-7 digital camera (Sony Corporation of Hong Kong Limited, Hong Kong). Morphological details were imaged using a Nikon SMZ1500 stereomicroscope equipped with a Nikon DS-Fi1 digital camera (Nikon, Tokyo, Japan), and surface details impressed on sediments were observed using a TESCAN MAIA3 scanning electron microscope (SEM) at Nanjing Institute of Geology and Palaeontology (Nanjing, China), CAS (NIGPAS). All figures were organised using Photoshop 7.0 for publication.

## 3. Results

**Order:** incertae sedis.

**Family:** incertae sedis.

**Genus:** *Archaebuda* Chen and Wang, emended.

**Emended generic diagnosis**: The flower bud is long-stalked. The stalk is straight, bearing scaly leaves. The scaly leaves are long, triangular and spirally arranged along a stalk. The bud is elongated and oval, including two types of foliar appendages. Foliar appendage type I is smaller, keeled, and round-tipped. Foliar appendage type II is bigger, keel-free, papery, notched at the tip, overlap with each other, and is of at least three layers. Gynoecium and possible androecium are surrounded by foliar appendage type II.

**Type species**: *Archaebuda lingyuanensis* Chen and Wang, 2022 [35].

**Species:** *Archaebuda cretaceae* sp. nov.

**Species diagnosis:** (in addition to that of the genus), gynoecium bottom-positioned.

**Description:** The specimen is a compression, 20 mm long and 9 mm wide, where most of the organic materials are missing, leaving impressions on two facing parts of siltstone (Figure 2a, Figure 3a and Appendix A). Foliar appendages of type I are at the base of the bud, smaller than the foliar appendages of type II, 2.9–3.5 mm long and 1.1–1.2 mm wide (Figure 2a, Figure 3a and Appendix A). Foliar appendages of type II are bigger than the foliar appendage of type I, 6.4–7 mm long and 3.6–4.6 mm wide, keel-free, papery, notched at the tip, with multiple layers overlapping each other and a longitudinally oriented texture (Figure 2a,c, Figure 3a,c, Appendix A). Possible traces of androecium could be seen, but no in situ pollen grains were recognised using a SEM (Figure 2a, Figure 3a and Appendix A). The gynoecium is centrally bottom-positioned in the flower bud, only 3 mm long and 1.4 mm wide, secluded by an ovary wall that is broken (Figure 2a,b, Figure 3a,b,d and Appendix A).

**Etymology**: *cretaceae*, for the Cretaceous, the age of the fossil.

**Holotype**: BMM79926 (Figure 2, Figure 3 and Appendix A).

**Type locality**: Dawangzhangzi, Lingyuan, Liaoning, China (41°15′N, 119°15′E, Figure 1).

**Type horizon and age**: the Yixian Formation, equivalent to the Barremian–Aptian, Early Cretaceous (approximately 125 Ma).

**Depository**: Blue Miracle Museum Science Research Studio, Guangzhou, China.

**Remarks**: The general appearance and dimensions of *Archaebuda cretaceae* are comparable to those of the type species *A. lingyuanensis* [35] (Table 1); therefore, these features allow us to place our new species in the genus *Archaebuda*. Different from the type species *Archaebuda lingyuanensis* [35], this new species has internal details, gynoecium (and possible androecium), preserved. Someone may be hesitant to accept the interpretation given by Chen and Wang [35], considering they could not convincingly exclude the possibility of the fossil being certain kinds of cones as they could neither confirm nor reject the existence of a cone axis in their fossil. A cone usually has a cone axis, which is now shown to be missing in *Archaebuda cretaceae*, excluding any possibility of *Archaebuda* being a cone. These additional details confirm the flower bud identity claimed for the type species *Archaebuda lingyuanensis* Chen and Wang [35], and our new species, *Archaebuda cretaceae*. Due to different preservation statuses in *Archaebuda lingyuanensis* Chen and Wang, and *Archaebuda cretaceae*, we currently cannot either confirm these two species as the same species or separate them as two species with confidence. The new species proposed here is only an expedited solution due to the limited data, and we look forward to further palaeobotanical progress that will shed new light on this issue.

The taxonomic position of *Archaebuda cretaceae* within angiosperms (such as order and family) is left open due to (1) the limited number of morphological characters and a (2) limited number of fossil specimens.

## 4. Discussions

The great diversity of angiosperms in the Yixian Formation is well known, including various reproductive organs of angiosperms [11,12,14,15,16,22,35]. However, among these fossils, flower buds are a rarity: there is only one previous record, *Archaebuda lingyuanensis* [35]. Such a sparse record of flower buds not only restricts us from better understanding of flower evolution but also reduces the credibility of interpreting such a specimen as a flower bud.

It is well known that previously reported reproductive organs of fossil angiosperms from the Yixian Formation showed little resemblance to extant flowers as they usually lacked the perianth-like part that is frequently seen in extant flowers. Their unusual morphologies make it hard to convince the general public of the truthful occurrence of Cretaceous fossil flowers. To make the situation worse, limited by the techniques extracting fossil specimens out of sediments and based on their own data, some palaeobotanists even hypothesised that early flowers are of small size (0.5 to 5 mm), although their fossils are of much younger ages [19,20]. Such an image of early flowers unavoidably leads to a scenario of flower evolution, namely big flowers seen in extant and fossil angiosperms, which are derived from these small flowers through various modifications. However, a caveat in mesofossil studies is that megafossils of fragile big flowers like *Archaefructus* would have been destroyed completely during the mesofossil sieving process, a process that targets the mesofossil only. The composite effects of the above two factors (naked early flowers and the overwhelming number of mesofossil flowers) have heavily misled botanists and prevented us from correctly understanding and interpreting character polarity in flower evolution. It is noteworthy that the occurrence of both *Archaebuda lingyuanensis* and *A. cretaceae* in the same outcrop of the Yixian Formation is significant in term of flower evolution. It appears that the former thought (early angiosperms have smaller flowers) is closely hinged with the method applied in mesofossil studies. The discoveries of large-sized *Archaebuda lingyuanensis* and *Archaebuda cretaceae* from the Early Cretaceous clearly undermine the validity of this view on flower evolution and may help to correct this biased view.

In light of the confirmed occurrence of big flower buds in the Early Cretaceous, the picture of flower evolution needs to be updated so it becomes superfluous to derive big flowers like *Archaebuda* from either smaller flowers of younger ages or coeval naked flowers like *Archaefructus*. The diversity of reproductive organs of angiosperms (flowers) in the Yixian Formation deserves our further investigation: Why are they so diverse? Do they have more ancient ancestors? What do their ancestors look like? We do not think that the situation in the Early Cretaceous Yixian Flora constitutes another “abominable puzzle”. Instead, there must be unrevealed truth embedded in the sediments of earlier ages (such as the Jurassic or earlier ages). As for the provenance of big flowers like *Archaebuda*, unlike what was thought formerly, maybe we should focus on the Jurassic, a period in which angiosperms or angiosperm-related taxa have been repeatedly reported [22].

Hitherto, there are only two individual reports of fossil flower buds. One of them, *Florigerminis* [34], was uncovered in the Middle–Late Jurassic Jiulongshan Formation of Inner Mongolia, China. This flower bud is physically connected with leafy branches and fruit. These physically connected organs demonstrate a phenomenon rarely seen in the fossil records, namely, flower buds and fruits in different developmental stages in a single specimen. The other is a congeneric species of *Archaebuda cretaceae*, *A. lingyuanensis* Chen and Wang, from exactly the same outcrop of the Yixian Formation [35]. This fossil was identified as a fossil flower bud based on its general morphology, three overlapping petals, the arrangement of the petals and their textures [35]. Unfortunately, the supposed androecium and gynoecium could not be verified in *Archaebuda lingyuanensis*. This shortcoming of the paper more or less reduces the credibility of the claim made by Chen and Wang [35]. Now, taking the discovery of coeval *Archaebuda cretaceae* with gynoecium and possible androecium into consideration, it becomes more convincing that flower buds did exist in the Early Cretaceous, as both androecium and gynoecium, which were impossible to be revealed in *Archaebuda lingyuanensis*, have now been verified in its congeneric species, *Archaebuda cretaceae*.

Compared to the Jurassic flower bud *Florigerminis* [34], two species of *Archaebuda* uncovered in the Early Cretaceous are much bigger in size. The flower bud of *Florigerminis* is only 3.8 mm long and 3.3 mm in diameter [34], smaller than *Archaebuda*, which is about 20 mm long and 9 mm in diameter. It is noteworthy that, although the fruit of *Florigerminis* is 11.5 mm long and 7.7 mm in diameter (much bigger than its own flower bud) [34], it is still smaller than both species of *Archaebuda*. These comparisons indicate that the stark size contrast between *Florigerminis* and *Archaebuda* cannot be attributed to different development stages. This big size of *Archaebuda* makes their flowers more conspicuous, although, currently, we have no idea how bright or colourful these flower buds may be. Conspicuous petals (corolla) are frequently seen in extant angiosperm flowers, and they are thought to be hinged with successful insect pollination in angiosperms. It is imaginable that big petals in *Archaebuda* are conducive to enhanced successful rate of insect pollination, especially considering there were fewer flowers with perianth or corolla in the Early Cretaceous. This conclusion implies that the ecological ties between flowers and insects (or other animals) may have been established as early as the Early Cretaceous, although we currently cannot estimate how old the earliest tie may be.

In previous studies, ovule-protecting is frequently taken as a feature unique to angiosperms. However, it appears that the function of perianth, petals or corolla is usually less emphasised or focused on. Gynoecium and androecium usually are very fragile and actively developing parts in flowers, and they may well become the favourite food candidates for insects or other animals. Therefore, protection for androecium and gynoecium against potential possible attackers provided by petals or corolla is a clear advantage in angiosperms’ survival struggle. We hope future work may shed more light on this issue.

## 5. Perspectives

Flower buds were rarely reported in previous studies, probably due to their lower potential to be fossilised. However, it appears that such a lack of reports on flower buds in the fossil record is hinged on the methods applied in palaeobotanical practice. As more digging is carried out and new technologies (including Micro-CT) are applied in palaeobotany, we have more power to extract information about fossil flowers embedded in sediments. This will, in turn, enhance our knowledge of fossil flowers and our perspective on angiosperm evolution in general. However, it should be borne in mind that a huge amount of data generated this way may over-dominate and mislead our thinking in some way, as occurred for mesofossil sieving (see above). So, we should try our best to apply as many technologies as possible in palaeobotany, calibrating their outcomes with each other to balance our view.

## 6. Conclusions

Discoveries of two congeneric species of flower buds, *Archaebuda lingyuanensis* and *A. cretaceae*, from the same outcrop of the Yixian Formation, indicate that angiosperms in the Early Cretaceous had big flowers that are frequently seen in extant angiosperms; such big-sized flowers were formerly under-studied. Together with the atypical flowers (naked ones), the occurrence of *Archaebuda* in the Yixian Formation indicates that angiosperms and flowers demonstrate a wider spectrum of diversified morphology 125 Ma ago. Big flowers with conspicuous petals apparently contribute to the success of some angiosperms in their early radiation and imply that they started their ecological interaction with animals (especially insects). We expect future studies will uncover more flower buds or other fossils of angiosperms, which will pave a way leading us to a better understanding of the evolution of early angiosperms.

## Figures and Tables

**Figure 1 biology-13-00413-f001:**
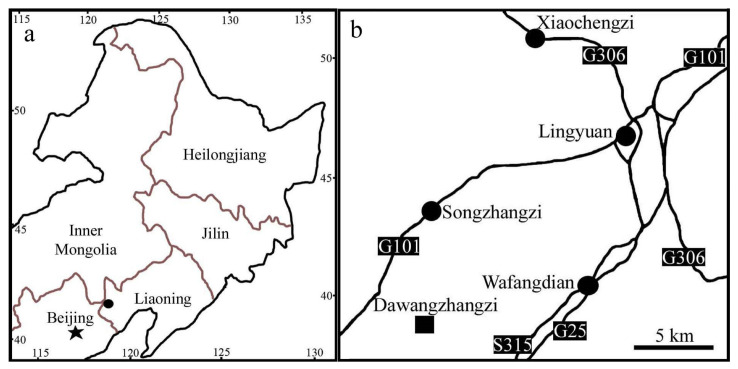
The fossil locality of *Archaebuda cretaceae*, Dawangzhangzi, Lingyuan, Liaoning, China. Modified from Han et al. [51], with permission and courtesy of Acta Geologica Sinica (English edition). (**a**) Fossil locality (black dot) in northeastern China. (**b**) Detailed position of fossil locality (square) in the suburb of the City of Lingyuan, Liaoning, China.

**Figure 2 biology-13-00413-f002:**
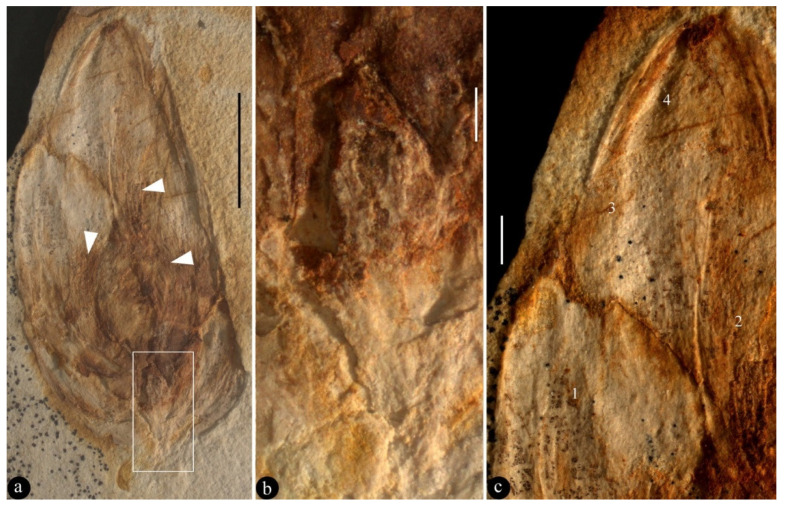
The holotype of *Archaebuda cretaceae* and its details. (**a**) The general view of the specimen. The possible androecium is marked with white triangles. Scale bar = 5 mm. (**b**) A detailed view of the rectangle is shown in Figure 2a, showing the gynoecium. Scale bar = 0.5 mm. (**c**) A detailed view of the tip portion of the bud shown in Figure 2a, showing different textures on and borders between different petals (1–4). Scale bar = 1 mm.

**Figure 3 biology-13-00413-f003:**
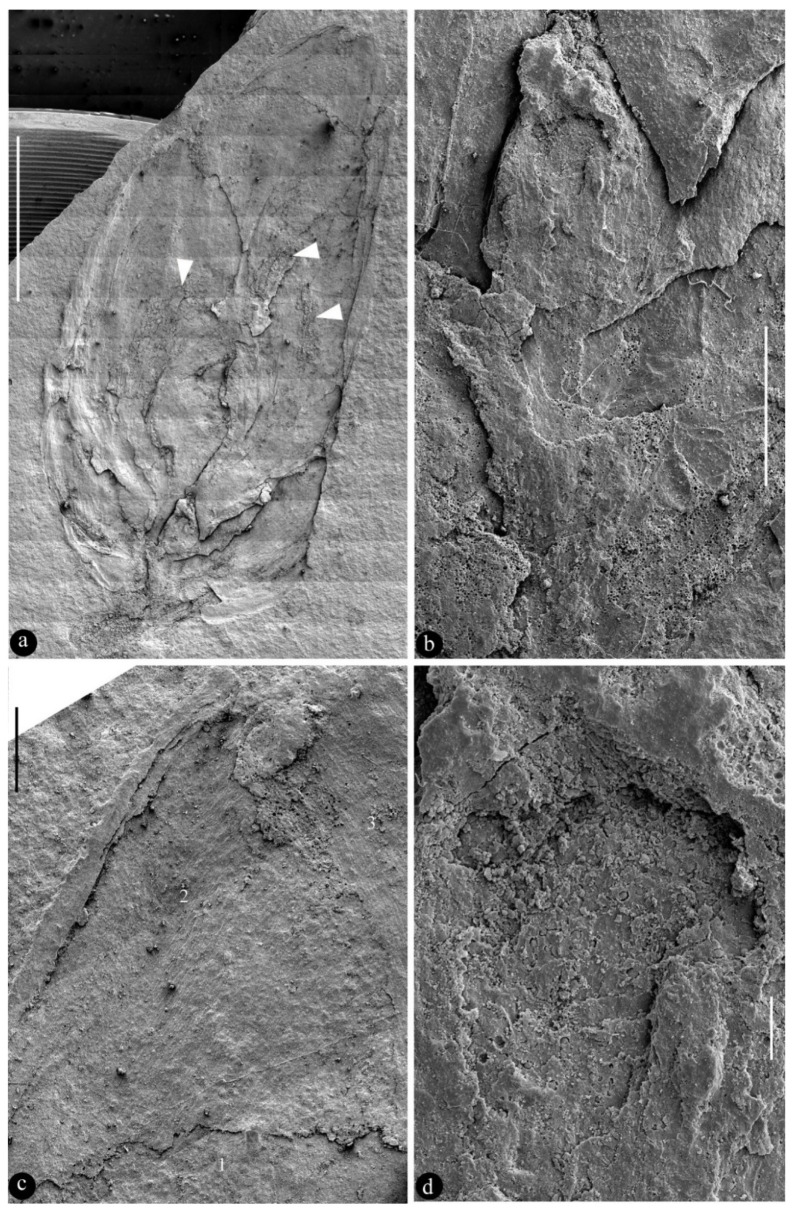
SEM of *Archaebuda cretaceae*. (**a**) General view of the specimen. The possible androecium is marked with white triangles. Scale bar = 5 mm. (**b**) A detailed view of the gynoecium, enlarged from the bottom of Figure 3a. Scale bar = 1 mm. (**c**) A detailed view of the tip of the flower bud shown in Figure 3a. Note the textures on different petals (1–3). Scale bar = 0.5 mm. (**d**) A detailed view of the gynoecium shown in Figure 3b, showing a broken ovary wall and exposed internal details in the gynoecium. Scale bar = 0.1 mm.

**Table 1 biology-13-00413-t001:** Comparison between *Archaebuda lingyuanensis* and *A. cretaceae*.

	*Archaebuda lingyuanensis*	*Archaebuda cretaceae*
Stalk	15 mm long	Not preserved
Length of bud	16 mm	20 mm
Width of bud	9 mm	9 mm
Foliar appendages type I, width	3 mm	1.1–1.2 mm
Foliar appendages type I, length	2.6 mm	2.9–3.5 mm
Foliar appendages type II, width	4.1–6.7 mm	3.6–4.6 mm
Foliar appendages type II, length	4.8–16.8 mm	6.4–18 mm
Androecium	Not visible	Possible
Gynoecium	Not visible	Present

## Data Availability

Data are contained within the article and its Appendix A.

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
