# Peer review of "Flower Buds Confirmed in the Early Cretaceous of China"

_biology, 2024, doi:10.3390/biology13060413_

Round 1

Reviewer 1 Report

Comments and Suggestions for Authors

1. The title of the manuscript could be changed. Don't start with 'internal details'. It doesn't seem to be interesting for a reader.

2. Introduction: The language is not well written. The introduction could be expanded with more details. The research gap and how it would be solved is not explained well. Why studying lower cretaceous and its significance is unclear in the introduction. Suggestion is to rewrite the introduction part.

3. References: All the references are stacked together. They could be distributed throughout the manuscript.

4. Discussion: The language of the discussion is not well written. There is no flow in this discussion. Is there a need to gather more data to come to final conclusions? This needs explanation.

5. Conclusion: Expand it in detail. There are no future prospectives to this research study. The significance of this study is not explained well. Kindly revise. 

6. The term 'Simple Summary' doesn't sound good to me. Not necessary as such if you have an abstract already. 

Overall, good research but more work is needed on this manuscript.

Comments on the Quality of English Language

There are a few grammatical errors in the manuscript. Certain sentences need to be changed for better understanding. 

Reviewer 2 Report

Comments and Suggestions for Authors

This short manuscript describes a possible new fossil flower from the Lower Cretaceous Yixian Formation of China. First of all, I must say that I am not a paleobotanist specialized in angiosperms. But I have serious doubts whether this specimen can be identified as a fossil flower bud. And if it is really a new fossil species, the justification must be more solid and comparisons with other angiosperms and non-angiosperm genera must be expanded. Please see the attached pdf with my comments for more details.

Comments on the Quality of English Language

English is not my native language, but I think the manuscript should be checked again.

Reviewer 3 Report

Comments and Suggestions for Authors

A newly discovered specimen, Archaebuda cretaca, collected from the Yixian Formation, presents a fossil flower bud with internal details. The genus Archaebuda, as emended by Chen and Wang, is characterized by elongated oval flower buds with long-stalked, straight stems bearing scaly leaves. These leaves are spirally arranged along the stalk, with two types of foliar appendages: type I, smaller and keeled with round tips, and type II, larger, keel-free, papery, and notched at the tip. This finding contributes to our understanding of early angiosperm evolution.

Major concerns:

The study introduces a new species, Archaebuda cretaca sp. nov., but does not elaborate on the criteria used for taxonomic classification. Details on morphological features, comparisons with known species, and justification for the assignment of a new taxon missing.

Detailed interpretations of the internal structures observed in the fossil missing.

authors briefly compared the newly discovered species with the type species A. lingyuanensis but does not conduct a comprehensive comparative analysis. A more thorough comparison, including additional morphological traits and ecological considerations is required.

Evolutionary implications of this finding missing.

Comments on the Quality of English Language

Overall, the quality of English is good

Round 2

Reviewer 1 Report

Comments and Suggestions for Authors

The manuscript has been sufficiently improved. Some minor editing is required if possible. Details of Archaebuda (appearance) must be briefly mention in the abstract. The introduction could be expanded a little. The introduction starts with a controversial statement. The figures need formatting. A separate section on future potential would be appreciated. Overall, improved manuscript. The references are related to the manuscript.

Author Response

The manuscript has been sufficiently improved. Some minor editing is required if possible. Details of Archaebuda (appearance) must be briefly mention in the abstract. Thanks. We have fixed the problem according to this suggestion. The introduction could be expanded a little. Thanks. We have fixed the problem according to this suggestion. The introduction starts with a controversial statement. Thanks. We have paraphrased according to this suggestion.The figures need formatting. The meaning of this word is unclear. What should we do to make it “formatted”? A separate section on future potential would be appreciated. Overall, improved manuscript. Thanks. The references are related to the manuscript. We have re-worked on the references.

Reviewer 2 Report

Comments and Suggestions for Authors

I notice that most of my suggestions and comments have been included in the revised version. However, it seems that the authors exaggerate in citing themselves (with a self-citation of 30%.). This should be fixed before this manuscript is published.

Author Response

I notice that most of my suggestions and comments have been included in the revised version. However, it seems that the authors exaggerate in citing themselves (with a self-citation of 30%.). This should be fixed before this manuscript is published. Thanks for this reminder. We have reduced the number of self-references down to 5.

Reviewer 3 Report

Comments and Suggestions for Authors

The Word file uploaded by the authors is difficult to read. It would be helpful if they provided a separate file with clear explanations for each comment. Additionally, the authors' comment that they have expanded their data but haven't mentioned the line numbers is causing unnecessary back-and-forth and is a waste of the reviewers' time.

Author Response

The Word file uploaded by the authors is difficult to read. It would be helpful if they provided a separate file with clear explanations for each comment. For your information, we have submitted a PDF file with all changed marked. This should be better than a separate list, as the changes are placed in the context. Additionally, the authors' comment that they have expanded their data but haven't mentioned the line numbers is causing unnecessary back-and-forth and is a waste of the reviewers' time. Sorry, causing so much trouble for you.

Round 3

Reviewer 3 Report

Comments and Suggestions for Authors

accepted

Author Response

Thanks.
